# Preparation and Characterization of Synchronous Chemical Conversion Coating on 6061 Aluminum Alloy/7075 Aluminum Alloy/Galvanized Steel Substrates

**Wen Zhan** [1,2,*], **Xinxiang Li** [2], **Xuzhen Qian** [2], **Yingpeng Li** [1], **Yunhu Ding** [2], **Yunhe Zu** [2], **Fan Xie** [2] and **Feng Tian** [2]

[1] China Academy of Machinery Science and Technology Group Co., Ltd., Haixi (Fujian) Institute, Sanming 365500, China
[2] State Key Laboratory of Special Surface Protection Materials and Application Technology, Wuhan Research Institute of Materials Protection, Wuhan 430030, China
[*] Correspondence: zhanwen@rimp.com.cn

**Abstract:** This paper aimed to develop synchronous chemical conversion coating on multi-metal substrates with good corrosion resistance to meet the primer process of new energy light vehicle bodies. Titanium/zirconium-based chemical conversion coatings were prepared on 6061 aluminum alloy/7075 aluminum alloy/galvanized steel substrates. By measuring the open circuit potential (OCP), the formation of a muti-metal synchronous conversion coating can be roughly divided into three steps. Potentiodynamic polarization (PDP) and electrochemical impedance spectroscopy (EIS) techniques showed that the self-corrosion current density of the conversion coating decreased significantly while the resistance increased. The surface morphology and composition of the conversion coatings were observed by scanning electron microscope (SEM) and X-ray photoelectron spectroscopy (XPS). Additionally, the micro-zone characteristics of conversion coatings were analyzed by an electron probe microanalyzer (EPMA). The synchronous conversion coatings exhibit uniformity and relative smoothness. Additionally, a number of tiny cracks, pores, intermetallic compounds, enrichments and inclusions provide efficient active sites for the nucleation of chemical conversion. Consequently, in the synchronous conversion coating, the structure of aluminum alloy mainly consists of $Al_2O_3/TiO_2/ZrO_2/ZrF_4$, while the structure of conversion coating of galvanized steel contains $TiO_2/Fe_2O_3/ZrO_2$.

**Keywords:** synchronous conversion coating; aluminum alloy; galvanized steel; titanium/zirconium-based

## 1. Introduction

In order to achieve the goal of reducing fuel consumption, reducing environmental pollution and saving resources, the automobile industry is moving towards new-energy and lightweight direction development [1,2]. In recent years, the Model S sedan was developed and manufactured by Tesla, an American manufacturer of pure electric vehicles. It adopts an all-aluminum body with both lightweight and high-strength characteristics, including the bottom coating technology of aluminum alloy and high-strength steel materials, such as six series of aluminum materials for the front compartment of the roof and bottom, seven series of aluminum materials for the front bumper and five series of aluminum materials for the four-door bracket. Because the body uses a large number of aluminum alloy parts, if the traditional phosphating technology is applied to an all-aluminum body as the bottom coating pretreatment process [3–5], due to the entry of $Al^{3+}$, the phosphating liquid is rapidly poisoned, the phosphating coating generated cannot meet the needs of automobile corrosion resistance and paint adhesion. Various metal body bottom coating manufacturing technology has become the focus of many new energy vehicles lightweight body manufacturing demand brands.

Chemical conversion technology is widely used in the surface treatment of aluminum alloy due to the simple advantages of materials and manufacturing methods. Among them, hexavalent chromate chemical conversion technology has a stable process and excellent corrosion resistance to passivation coating. The coating layer is uniform and golden yellow, which is the most widely used process product in the existing market [6–9]. Nevertheless, hexavalent chromium is a highly toxic pollution substance and carcinogenicity, and the European Union issued a series of RoHS directives banning the use of hexavalent chromium earlier. Scholars have researched the substitution of the hexavalent chromium chemical oxidation method, titanium/zirconium conversion coating, rare earth conversion coating, permanganate conversion coating, vanadate conversion coating, silicone coating and other aspects [10–18]. Nordlien, J.H. et al. [19] discussed the characteristics of Alodine2840 on 6060 aluminum alloy chemical conversion coating. The presence of mixed oxides of Zr, Ti and Al in the passivated elements of the aluminum alloy after 30 s was tested. It is suggested that Ti/Zr oxide coatings grow preferentially around the intermetallic particles, which is beneficial in reducing the cathode reaction of grains. Lunder, O. et al. [20] focused on the effects of intermetallic particles, stirring and pH value on the formation and growth of Ti/Zr oxide coating on 6060 aluminum alloy. It was pointed out that the Ti/Zr oxide coating is composed of thin Ti in the far region of metal particles and the potential growth range of a-Al (Fe, Mn)Si phase particles. The cathodic reaction current changed very little with the treatment time, and the cathodic activity did not decrease with the coating, which was beneficial to the increase in the thickness of the chemical conversion coating. Andreatta, F. et al. [21] elaborated on the deposition behavior of Ti/Zr oxide coating at different growth stages, pointing out that the negative polarity uneven area of aluminum alloy is the driving force for the formation of Ti/Zr coating and $F^-$ can promote the natural formation of Ti/Zr coating at the early stage of coating formation. Hence, Ti/Zr oxide coating grows beside the negative polarity intermetallic particles. The conversion coating can reduce the matrix's potential difference and improve the aluminum alloy's corrosion resistance by covering the intermetallic region of opposing polarity. The phosphating technology of the high-strength steel body is mature and practical. Researchers have also carried out research on the chemical conversion coating and its corrosion mechanism instead of phosphating technology. Sarabi et al. [22] studied the coating formation behavior of Ti-based chemical conversion coating on a cold rolled steel plate. $Ni^{2+}$ can promote uniform and thick coating growth. The electrochemical test showed that TiNiCC coating takes significant polarization resistance and low corrosion current. TiMoCC coating generated by adding Molybdenum salt has a network structure with many cracks and relatively small polarization resistance, which is not conducive to improving corrosion resistance. The addition of phytic acid helps refine the deposited grain, smooth the surface, and has good bonding with the coating, but the corrosion resistance is not apparent. Zhang et al. [23] explored the growth model of lanthanum metal chemical conversion coating on galvanized steel sheets. They found that the lanthanum conversion coating first grew rapidly along the edge of the zinc grain. This region was formed first at a particular stage and gradually expanded to the whole surface.

Most scholars have generally recognized the influence of different intermetallic compounds on the growth mechanism of the chemical conversion coating of aluminum alloy [24–26], which is similar to the model of Cu-rich or Fe-rich cathode region proposed by previous scholars studying the chemical conversion mechanism of traditional chromite. There are few reports on the mechanism of synchronous chemical conversion of multi-metal surfaces. This paper identified the deposition and corrosion mechanism and the microzones structure evolution of multi-metal chemical conversion composed of six-series aluminum alloy/seven-series aluminum alloy and galvanized steel.

## 2. Materials and Methods

The 6061 aluminum alloy/7075 aluminum alloy/galvanized steel samples of 20 mm × 20 mm × 2 mm were prepared as substrate materials for investigation. After polished (employing 400-mesh, 1200-mesh and 2000-mesh sandpaper successively) and

acid degreasing treatment (including 5% ZHM-1026 production employed by Wuhan Research Institute of Materials Protection, Wuhan, China), the pretreated 6061, 7075 aluminum alloys and galvanized steel were immersed simultaneously into the conversion bath, which is made up of $H_2TiF_6$ 2.2 mL/L, $H_2ZrF_6$ 1 mL/L, adjusting solution pH 3.9 by ammonia and conversion temperature at 35 °C. Different conversion times (from 100 s to 140 s) were used to investigate the conversion coating process. Finally, the coated specimens were rinsed in deionized water, dried with air and used further for characterization.

Electrochemical studies such as potentiodynamic polarization (PDP) and electrochemical impedance spectroscopy (EIS) techniques were employed using the CHI760E electrochemical workstation supplied by Shanghai Chenhua Instruments Inc, Shanghai, China. The tests were carried out in 3.5 mass% NaCl solution for 25 min to obtain a steady open circuit potential (OCP), using the standard three-electrode cell equipped with the coated sample (1 cm × 1 cm) as a working electrode, platinum foil as the counter electrode, and a saturated calomel electrode (SCE) as the reference electrode. The initial and final potential of the polarization was set from −1.0 to −0.4 V, and the scanning speed was 0.001 V/s. It was remarked that potential scan rate has an important role in order to minimize the effects of distortion in Tafel slopes and corrosion current density analyses, as previously reported [27–29]. However, based on these reports, the adopted 0.001 mV/s has no deleterious effects on those Tafel extrapolations to determine the corrosion current densities of the examined samples. After the experiment, Tafel fitting was performed to obtain the self-corrosion potential ($E_{corr}$) and corrosion current density ($i_{corr}$). Electrochemical impedance spectra were obtained at the OCP in the frequency range from 10 kHz to 0.01 Hz, with 10 mV as the amplitude of the perturbation signal. The impedance data were displayed as Nyquist curves, which could establish the electrical equivalent circuits (EEC) by the fitting of ZSimPwin 3.10 (E Chem Software, Ann Arbor, MI, USA).

The surface morphology of the chemical conversion coatings on different metals was observed using Sigma 300 SEM (ZEISS, Oberkochen, Germany), and their chemical compositions were analyzed based on the energy spectrum (EDS). A Kratos AXIS SUPRA spectrometer (Al-K$\alpha$ = 1486.6 eV) equipped with an aluminum anode was used for qualitative analysis of the valence state and compound type of each element in the conversion coating. The obtained X-ray photoelectron spectroscopy (XPS) spectra were calibrated with the adventitious carbon peak (1 s) as a reference at 284.6 eV. The variation in the microstructure of the aluminum alloy in different coating formation stages was studied by the EPMA-1720H Electron Probe X-ray Microanalyzer (EPMA) of Shimadzu Corporation of Japan, Kyoto, Japan, and the coating formation process was inferred.

### 3. Results and Discussion

*3.1. OCP Measurements during the Formation of Conversion Coating*

The formation of synchronous chemical conversion was followed by measuring the OCP of 6061 aluminum alloy/7075 aluminum alloy/galvanized steel samples immersed in a conversion bath of $H_2TiF_6$ 2.2 mL·L$^{-1}$, $H_2ZrF_6$ 1 mL·L$^{-1}$ (Figure 1). The OCP test is carried out in the conversion solution, and the formation process of the conversion film is judged by its potential change. There are three distinct phases of all coating formation. (i) The natural multi-metal oxide film is thinned by the attack by F$^-$ ions, and the surface is activated for coating formation. The OCP of 6061/7075 aluminum alloy/galvanized steel samples dropped rapidly to more negative values within the first 93.4 s. At this stage, the surface of the sample begins to nucleate, but the film formation rate is far less than the dissolution rate of natural oxide film, so the potential drops rapidly. (ii) After the removal of natural multi-metal oxide, hydrogen evolution and oxygen reduction reactions, due to the increase in local alkalinity, synchronous chemical conversion coatings start to form on cathodic areas of the substrate, afterward growing in a lateral direction until completely covering the substrate. The maximum in the OCP curves (the OCP of 6061 aluminum alloy/7075 aluminum alloy/galvanized steel samples increased to 805.2 s/119.9 s/120.1 s, respectively) were taken as the optimal conversion time, at which the coating is fully formed.

Furthermore, the position of optimal conversion time could be checked previously by measuring electrochemical properties and analyzing the morphology surface by SEM/EDS of samples prepared at different synchronous conversion times. (iii) The complete formation of synchronous chemical conversion coatings, when the potential plateau is established after 200 s, the coatings tend to grow further. However, due to internal stress and the formation of hydrogen microbubbles or pores, it cracks and loses anticorrosion properties.

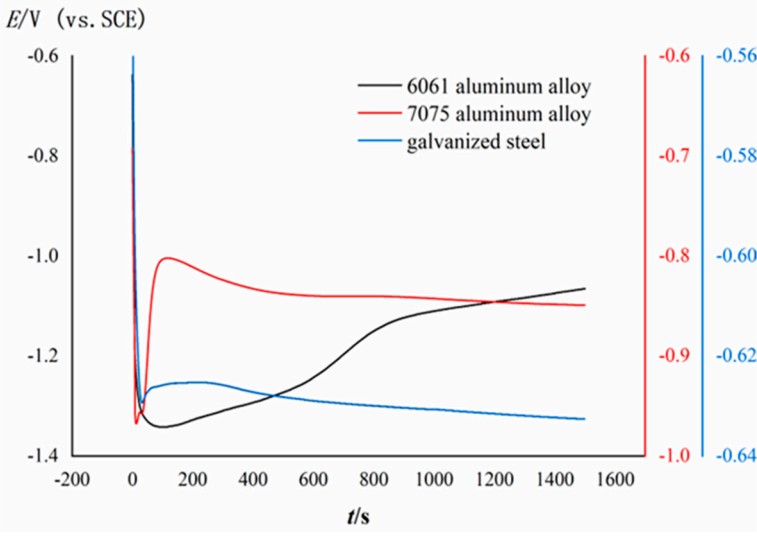

**Figure 1.** Open circuit potential during the formation of synchronous chemical conversion coatings of multi-metal.

### 3.2. Electrochemical Analysis

PDP and EIS techniques were utilized to investigate the electrochemical behavior without and with the synchronous chemical conversion of multi-metal in 3.5 wt% NaCl solution at conversion times 120 s are shown in Figure 2. The EEC of the 6061 aluminum alloy/7075 aluminum alloy/galvanized steel samples was curved fit for the obtained plots in Figure 3. Since an equivalent circuit was used (as shown in Figure 3) in order to determine the simulated values and compare them with experimental data, a CNLS (complex non-linear least squares) simulation was used, as previously reported [27,30–33]. In the EEC models, $R_s$ presents the electrolyte solution resistance between the sample and reference electrode, and $R_f$ refers to the access of the electrolyte to the alloy through pores and cracks of partial protective coating in parallel with $Q_1$, which is a constant phase element related with the entire barrier layer. $R_{ct}$ is the charge transfer resistance showing the protective properties of the coating, while $Q_2$ is the capacitance related to the electrical double layer. The values of the Tafel and EEC analysis are given in Table 1. The relevant analysis of the Bode and Bode Phase is shown in the Supplementary Materials Figure S1. Bode and Bode Phase graphs.

Compared with the untreated blank sample, the $i_{corr}$ of 6061 aluminum alloy synchronous conversion coating decreased from 1.096 µA/cm$^2$ to 0.174 µA/cm$^2$, and the $E_{corr}$ was positively shifted from −1.151 to −1.084 V. After EIS fitting, the $R_{ct}$ of the blank sample is only 0.268 Ω, while the $R_{ct}$ of conversion coating resistance suddenly increases to 846,10 Ω. In general, the greater the film resistance, the stronger the corrosion resistance of the conversion coating [34,35]. This is consistent with the previous Tafel polarization curve experimental results. The $i_{corr}$ of 7075 aluminum alloy decreases from 1.470 µA/cm$^2$ without conversion sample to 0.018 µA/cm$^2$ with the synchronous chemical conversion, and the film resistance increases from 19,330 to 23,250 Ω, respectively. The results show that the formation of synchronous chemical conversion coating can improve the corrosion resistance of 7075 aluminum alloy to a certain extent, but the $R_{ct}$ is not significantly increased due to there being some copper-enriched phases in samples [36]. At the same time, the $i_{corr}$ of galvanized steel without a conversion sample decreases from 6.312 µA/cm$^2$ to

1.012 µA/cm$^2$ after the chemical conversion treatment. The self-corrosion current density decreased significantly, which was only 1/6 of the former. The E$_{corr}$ changes from −0.938 to −0.838 V, and the film resistance increases from 5441 to 13,010 Ω, respectively. Compared with aluminum alloy, the improvement degree of these electrochemical performance indexes is slightly lower. This may be affected by the intrinsic characteristics of the galvanized steel substrate and the based system effect on the conversion coating. All these indicate that the corrosion resistance of 6061 aluminum alloy/7075 aluminum alloy/galvanized steel was enhanced at different levels after synchronous chemical conversion.

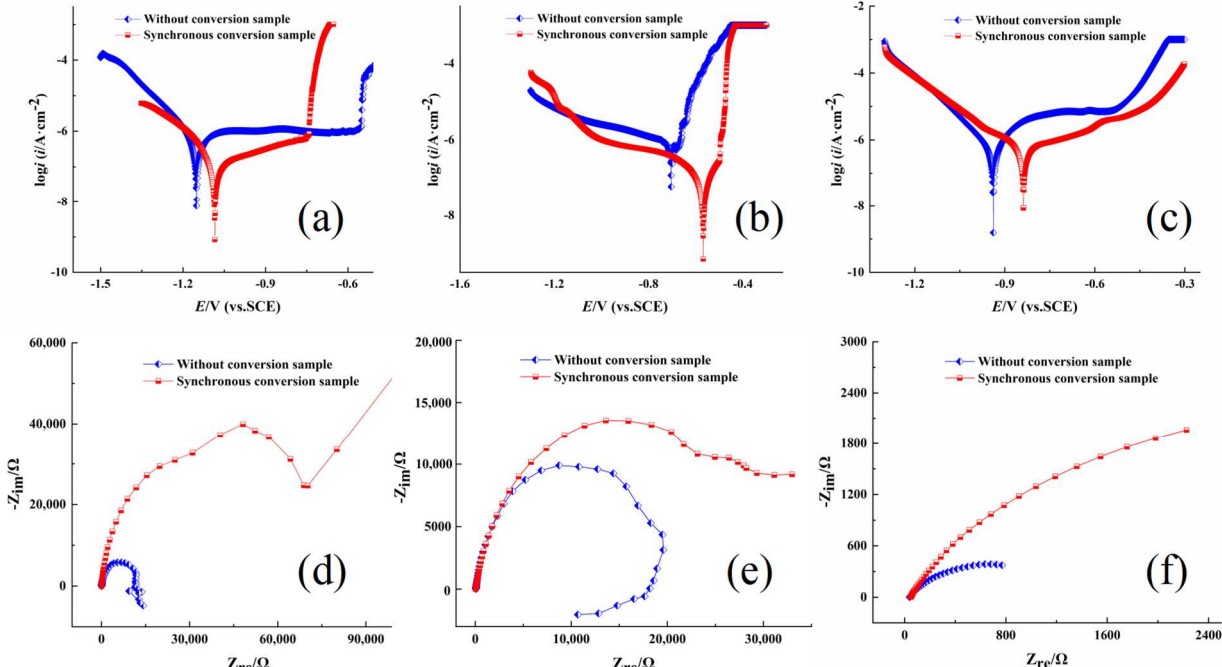

**Figure 2.** Electrochemical analysis for without and with synchronous chemical conversion coatings of multi−metal samples: (**a**) Polarization curves of AA6061, (**b**) polarization curves of AA7075, (**c**) polarization curves of galvanized steel, (**d**) Nyquist plots of AA6061, (**e**) Nyquist plots of AA7075, (**f**) Nyquist plots of galvanized steel.

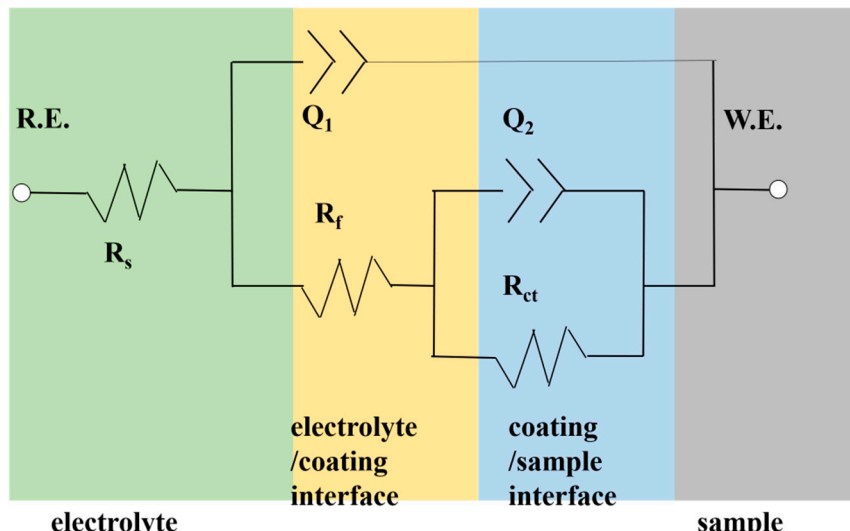

**Figure 3.** Equivalent circuits for untreated and simultaneously chemical-converted treated multi-metal samples.

**Table 1.** The values of the Tafel and EEC analysis for without and with the synchronous chemical conversion of multi-metal samples.

| Samples | $i_{corr}/\mu A \cdot cm^{-2}$ | $E_{corr}/V$ | $R_{ct}/\Omega$ | $R_s/\Omega$ | $R_f/\Omega$ | $Q_1$ | $Q_2$ |
|---|---|---|---|---|---|---|---|
| with conversion AA6061 | 0.174 | −1.084 | 84610 | 8.66 | 3675 | $6.317 \times 10^{-5}$ | $9.334 \times 10^{-5}$ |
| without conversion AA6061 | 1.096 | −1.151 | 0.268 | 12.97 | 919 | $8.661 \times 10^{-4}$ | $6.221 \times 10^{-4}$ |
| with conversion AA7075 | 0.018 | −0.603 | 23250 | 7.03 | 2876 | $9.112 \times 10^{-5}$ | $3.666 \times 10^{-5}$ |
| without conversion AA7075 | 1.470 | −0.653 | 19330 | 11.64 | 799 | $7.663 \times 10^{-4}$ | $2.729 \times 10^{-4}$ |
| with conversion galvanized steel | 1.012 | −0.838 | 13010 | 6.07 | 2017 | $1.117 \times 10^{-4}$ | $3.915 \times 10^{-4}$ |
| without conversion galvanized steel | 6.312 | −0.938 | 5441 | 9.88 | 366 | $7.664 \times 10^{-4}$ | $9.226 \times 10^{-3}$ |

### 3.3. Surface Characterization

Figure 4 shows the morphology of 6061 aluminum alloy samples with synchronous chemical conversion at different treatment times. At 100 s (Figure 4a), there are many tiny pores on the surface of 6061 aluminum alloy, which is mainly due to the hydrogen evolution of the cathode reaction at the early stage of film formation. After 110 s (Figure 4b–e), the pores were significantly reduced, and white granular materials with a size of 1–3 µm appeared inside and outside the pores, which were mainly intermetallic compounds (IMC). It is mainly composed of the α-Al (Fe, Mn)Si secondary phase [37], which provides the cathode potential for chemical transformation and promotes the uniformity and integrity of the transformation film. Due to the dominant factor of coating growth, rapid deposition leads to the film stacking phenomenon and local cracks in chemical conversion deposition. The distribution of Ti and Zr elements are small white aggregates in the pores (Figure 4f), and the distribution of Fe elements and Ti/Zr are mostly overlapped, which verifies the rule that the deposition of chemical conversion is preferentially on intermetallic compounds.

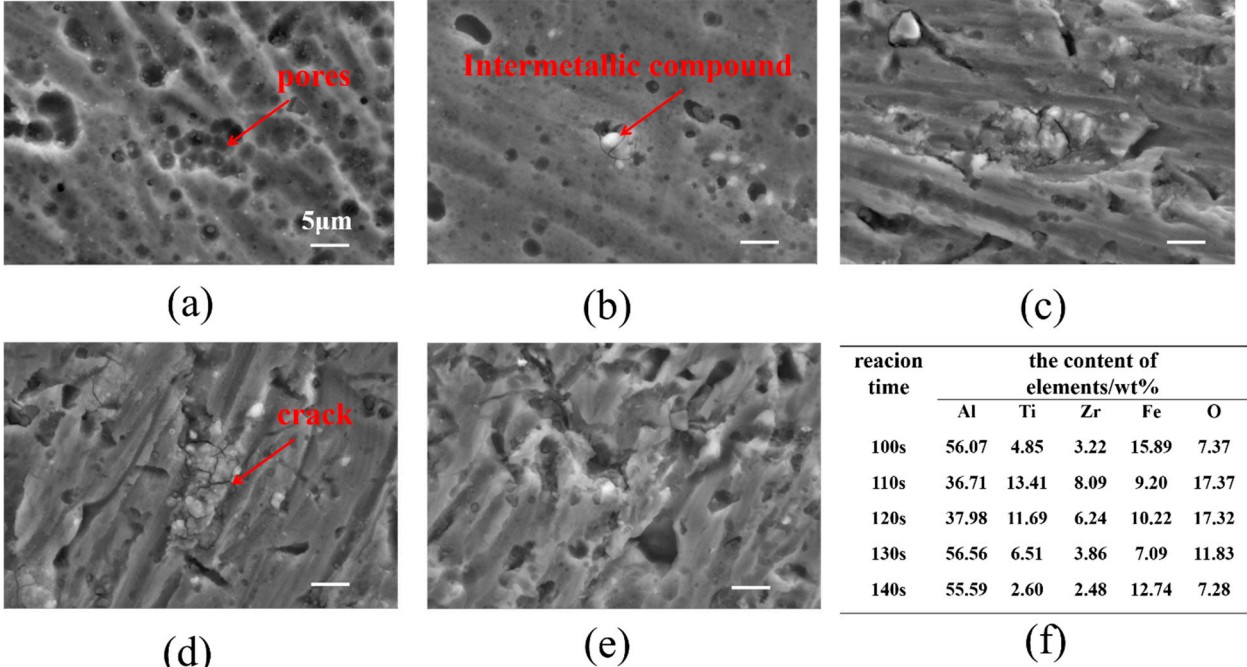

**Figure 4.** Results of SEM and EDS of 6061 aluminum alloy with synchronous chemical conversion at different times: (**a**) at 100 s, (**b**) at 110 s, (**c**) at 120 s, (**d**) at 130 s, (**e**) at 140 s, (**f**) the EDS result of the conversion coating.

Figure 5 shows the uniform and dense morphology of 7075 aluminum alloy samples with synchronous chemical conversion at different conversion treatment times. Additionally, there are also some tiny cracks and pores after 110 s (Figure 5b–e), and the cracks occurred on the local accumulation in the vicinity of the grain boundaries. A small amount

of white granular material with a 1 μm particle size distributed around it represents the existence of IMC. The interlaced cracks contain white granular materials with a particle size of 3–5 μm, which are mainly Cu enrichment IMC [38,39]. There are many titanium or zirconium oxides around IMC and enrichment, which provides an effective active site for nucleation of chemical conversion. The appearance of grain boundary cracks is easy to cause intergranular corrosion. The existence of a copper enrichment phase in the grain boundary enhances the potential difference in this microzone to a certain extent, which is beneficial to the conversion and deposition of the chemical conversion coating. However, the grain boundary cracks were not improved obviously from the morphology of SEM. At a reaction time of 120 s, the concentrations of Ti and Zr reached 14.5 and 7.4 wt% (Figure 5f), which was identical to the formation stage of OCP measurements.

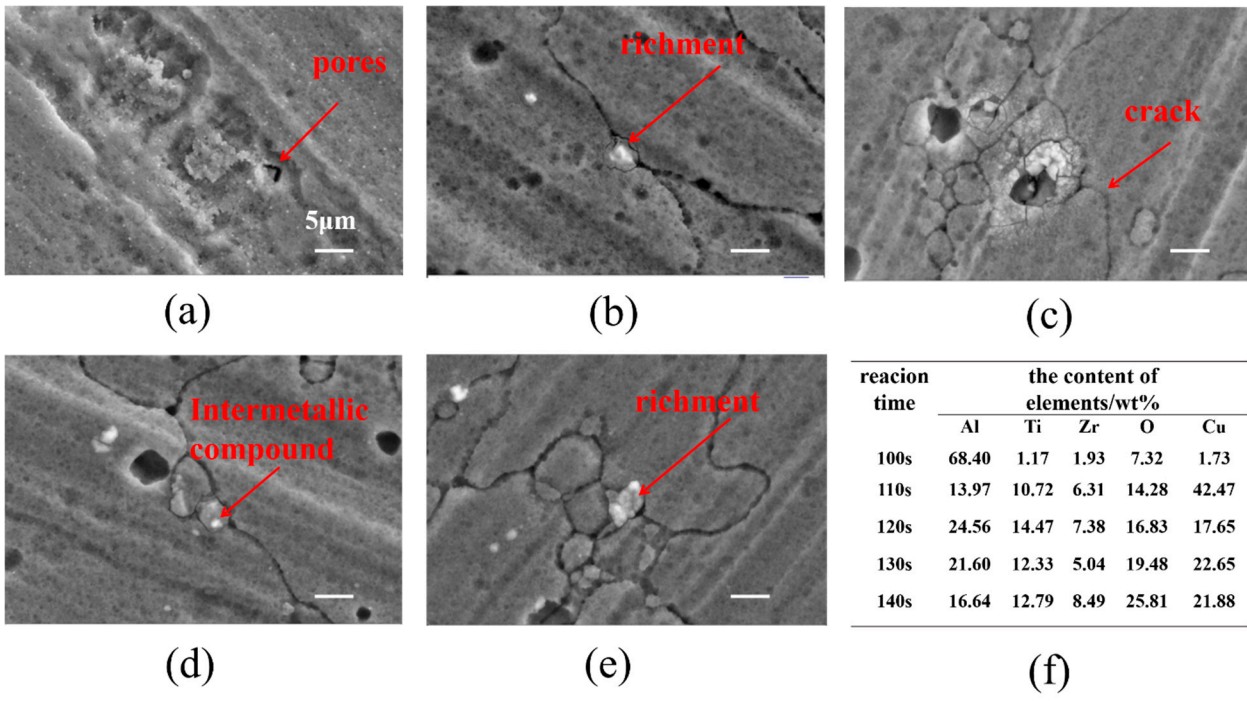

**Figure 5.** Results of SEM and EDS of 7075 aluminum alloy with synchronous chemical conversion at different times: (**a**) at 100 s, (**b**) at 110 s, (**c**) at 120 s, (**d**) at 130 s, (**e**) at 140 s, (**f**) the EDS result of the conversion coating.

The surface morphology of galvanized steel is shown to be uniform and relatively smooth, with few holes and no intermetallic compounds found on the surface (Figure 6). However, titanium or zirconium oxide film white granular material distribution in the protrusion or depression. When the conversion reaches 120 s (Figure 6c), cracks can be observed on the oxide surface distributed in the white block enrichment, mainly due to the uneven thickness caused by the excessive accumulation of film particles, resulting in stress fission. The coating has a few inclusions, including antimony/calcium/carbon oxide, which could be derived from the metallurgical process. By means of EDS (Figure 6f), it is speculated that the chemical conversion coating on the surface of galvanized steel is preferentially formed around the inclusion of carbon oxide, and the rest is distributed dispersively. In addition, the dissolution of $Al^{3+}$ in the two species of aluminum alloys reacted on the surface of galvanized steel to form a small amount of aluminide.

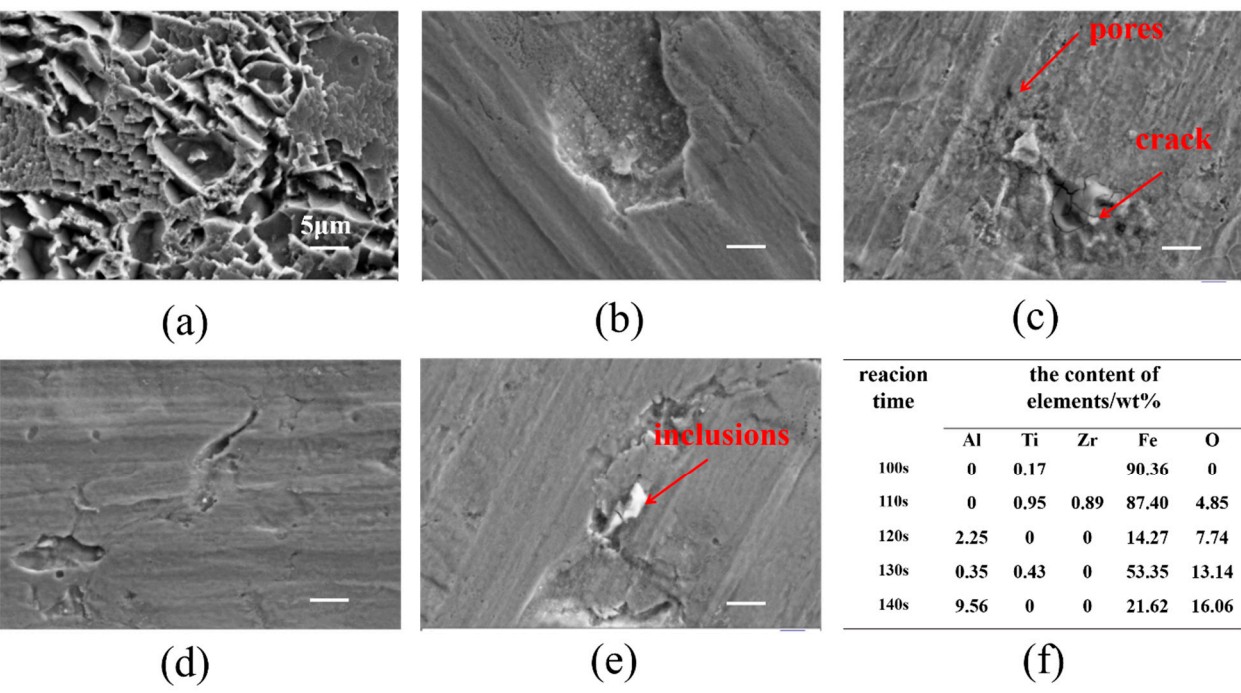

**Figure 6.** Results of SEM and EDS of galvanized steel with synchronous chemical conversion at different times: (**a**) at 100 s, (**b**) at 110 s, (**c**) at 120 s, (**d**) at 130 s, (**e**) at 140 s, (**f**) the EDS result of the conversion coating.

*3.4. EPMA Analysis*

In particular, intermetallic compounds, enrichment and inclusions have a certain effect on the growth of the chemical conversion coating. In order to further judge the influence of the main characteristics of various metal microzones on the conversion coating evolution, electron probe technology was used to investigate the element change rule of the microzone characteristic region under different conversion times.

Figure 7 shows the intermetallic compounds' characteristic region of 6061 aluminum alloy samples as the treatment time from 100 to 140 s. There are many elements, Ti and Zr, near the intermetallic compound, with Fe as the main component. With the increase in time, the amount of Ti and Zr shows a trend of increasing first and then decreasing, and the content of Ti on the intermetallic compound surface is higher than that of Zr. Similarly, the Cu-rich characteristic of 7075 aluminum alloy from 100 to 140 s was shown in Figure 8. The white spots and pores in the backscattering electron map mainly contain O, Cu, Ti and Zr elements, and the Cu-enriched phase easily occurs between the pores. Ti and Zr oxides are concentrated around the enrichment, increasing their contents and then decreasing them. It can be seen from Figure 9 that the microzone of galvanized steel appears bubble-like black substances, which are mainly composed of C, O, Ti and Zr elements. Due to its large particle size, it is speculated that the carbon oxides are inclusions precipitated by metallurgical processes. The Ti/Zr oxide deposited by chemical conversion is easy to grow on its surface and gradually accumulates and distributes diffusely. It was evidenced that the Ti/Zr conversion layers were firstly formed on the surface of cathodic α-Al (Fe, Mn) Si particles/Cu enrichment/carbon oxides inclusions and the surrounding areas, leading to a significant variation in the evolution of synchronous chemical coating conversion.

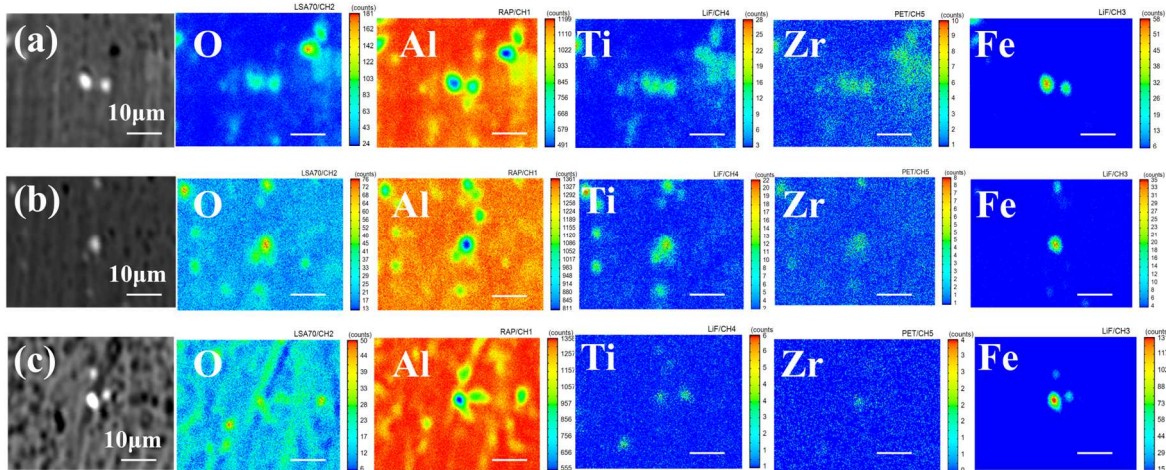

**Figure 7.** Results of EMPA of 6061 aluminum alloy with synchronous chemical conversion at different times: (**a**) at 100 s, (**b**) at 120 s, (**c**) at 140 s.

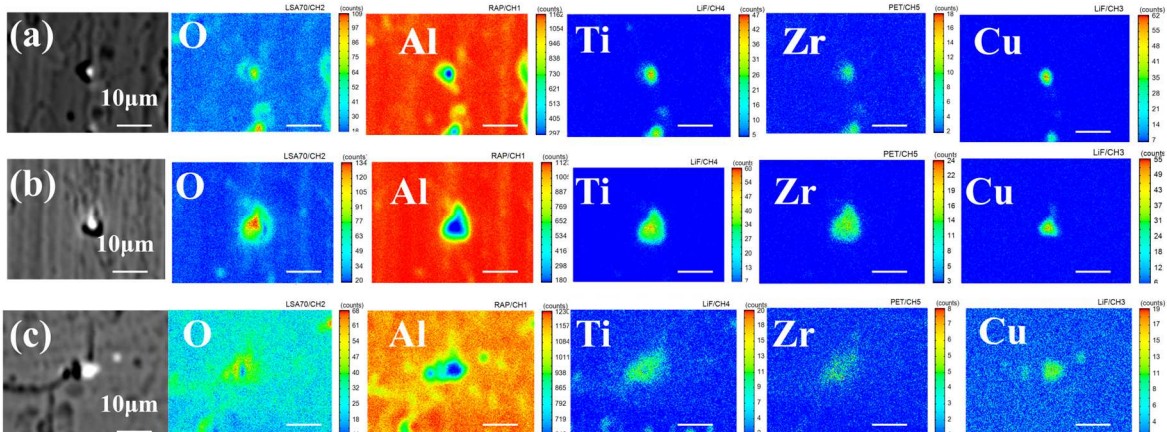

**Figure 8.** Results of EMPA of 7075 aluminum alloy with synchronous chemical conversion at different times: (**a**) at 100 s, (**b**) at 120 s, (**c**) at 140 s.

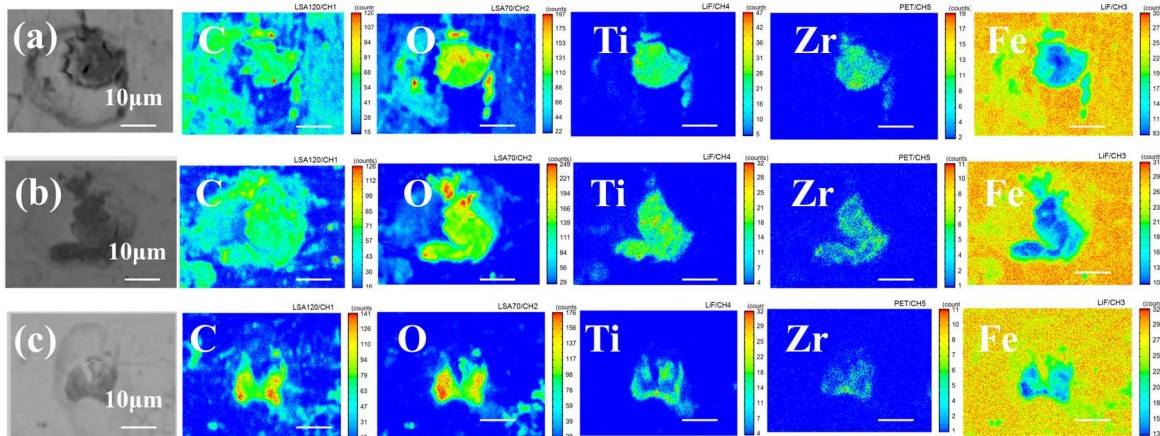

**Figure 9.** Results of EMPA of galvanized steel with synchronous chemical conversion at different times: (**a**) at 100 s, (**b**) at 120 s, (**c**) at 140 s.

*3.5. XPS Analysis*

The XPS spectra of synchronous chemical conversion coatings of 6061 aluminum alloy/7075 aluminum alloy/galvanized steel were recorded to characterize the chemical compositions shown in Figure 10, which indicated the existence of Al, Ti, Zr, O and F species

for 6061 aluminum alloy/7075 aluminum alloy conversion coatings, and the galvanized steel conversion coating was mainly composed of Ti, Zr, Fe, O and F elements. The binding energies of 458.3 and 464.5 eV in deconvolution of Ti 2p for both coatings represent the Ti 2p3/2 and Ti 2p1/2 of $TiO_2$, respectively. O1s spectrum can be fitted to two single peaks of O1s A (530.1 eV) and O1s B (531.9 eV), which belong to $TiO_2$ 1s and $Al_2O_3$ 1s, respectively. The Zr 3d is fitted with two peaks at 182.8 and 185.0 eV, which correspond to $ZrO_2$ and ZrF4 on 6061 aluminum alloy/7075 aluminum alloy coatings [40,41]. However, this signal could be assigned weakly to the galvanized steel species. F1s are fitted with two peaks at 684.5 and 685.1 eV, which correspond to $ZrF_4$ and $AlF_6^{3-}$ [42]. Al2p is fitted with two peaks at 74.5 and 74.76 eV, which correspond to $Al_2O_3$ [43]. Furthermore, Fe 2p is fitted with two peaks at 710.04 and 723.5 eV, which correspond to $Fe_2O_3$ and FeOOH [44]. The original peak of the Ca 2p spectrum contains a set of double peaks, and two satellite peaks belong to $CaCO_3$ from steel mixed on the surface. In general, the structure of the synchronous conversion coating of 6061 aluminum alloy and 7075 aluminum alloy mainly consist of $Al_2O_3/TiO_2/ZrO_2/ZrF_4$. The structure of the titanium–zirconium conversion coating of galvanized steel contains $TiO_2/Fe_2O_3/ZrO_2$.

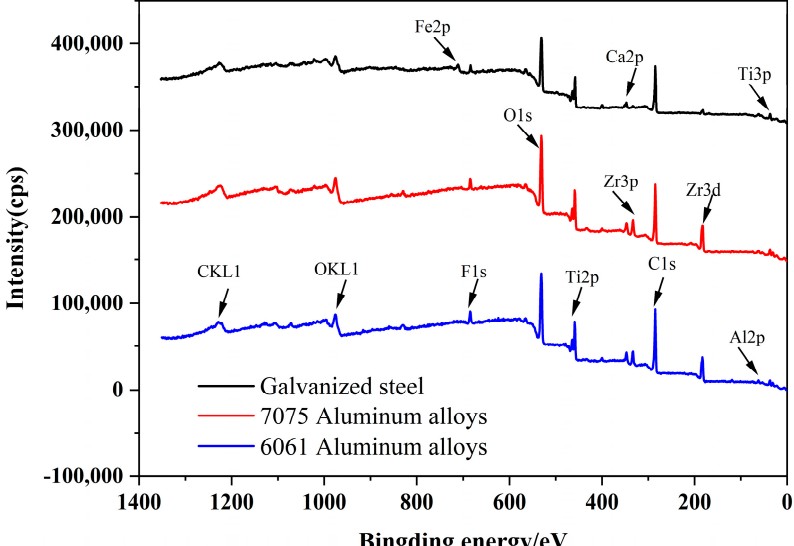

**Figure 10.** XPS spectra for synchronous chemical conversion coatings of 6061 aluminum alloy/7075 aluminum alloy/galvanized steel.

### 3.6. Discussion of the Formation of Synchronous Chemical Conversion Coatings

Through the above characterization and analysis of surface morphology, electron probe and microstructure, it can be seen that when a variety of aluminum alloys and galvanized steel are immersed in the conversion solution simultaneously, due to the electric potential difference between various metal elements and microzones characteristics, microcell reaction is easy to form on the surface of the substrate. The characteristics of micro cathode region, such as intermetallics, enrichment phases and inclusions, can especially drive the deposition of conversion coatings on and around its surface. According to the above results, we supposed that the framework of multi-metal synchronous conversion coatings could be roughly divided into three steps by the following reaction and schematic (Figure 11):

A. The dissolution of Al or Zn/Fe on the substrate surface in a conversion bath [45].

$$Al \rightarrow Al^{3+} + 3e^-$$

$$Zn \rightarrow Zn^{2+} + 2e^-$$

$$Fe \rightarrow Fe^{2+} + 2e^- \rightarrow Fe^{3+} + e^-$$

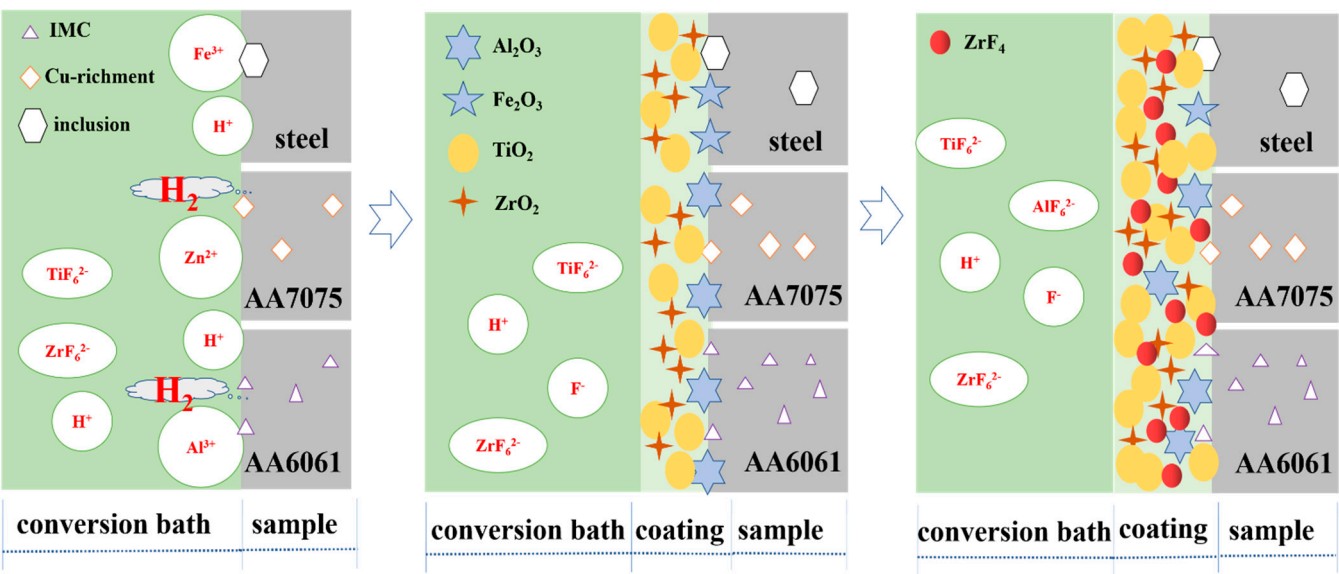

**Figure 11.** Schematic representations showing the evolution of the synchronous chemical conversion coatings on 6061 aluminum alloy/7075 aluminum alloy/galvanized steel.

B. The surface nucleates and grows at the characteristics of the microzones, forming conversion coatings and continuously depositing. Both cathodic reactions create local alkalinity adjacent to the microzones characteristics favoring the precipitation of $TiO_2$, $ZrO_2$, $Al_2O_3$ and $Fe_2O_3$ [46,47].

$$2H_2O + O_2(aq) + 4e^- \rightarrow 4OH^-(aq)$$

$$2H^+ + 2e^- \rightarrow H_2\uparrow$$

$$2Al^{3+} + 6OH^- \rightarrow Al_2O_3\downarrow + 3H_2O$$

$$2Fe^{3+} + 6OH^- \rightarrow Fe_2O_3\downarrow + 3H_2O$$

$$TiF_6{}^{2-} + 4OH^- \rightarrow TiO_2\downarrow + 2H_2O + 6F^-$$

$$ZrF_6{}^{2-} + 4OH^- \rightarrow ZrO_2\downarrow + 2H_2O + 6F^-$$

C. The dissolution of the micro anode and the formation of conversion coatings in the micro cathode region reach a dynamic equilibrium.

$$Al^{3+} + ZrF_6{}^{2-} \rightarrow AlF_6{}^{3-} + Zr^{4+} \tag{1}$$

$$Zr^{4+} + 4F^- \rightarrow ZrF_4 \tag{2}$$

At the last stage, the film-forming system is relatively complete, and the conversion coating is mainly composed of various metal oxides or fluorides [48]. The presence of microzones characteristics (intermetallics, enrichment phases and inclusions) than there are in the multi-metal will increase the rate of conversion coating growth, leading to a shorter optimal time of conversion. At the same time, the less stable natural metal oxide film will be easier to remove by the action of free fluoride ions, giving a faster rate of surface activation. If we continue to extend the conversion time, the film-forming of synchronous conversion coatings may produce cracks due to accumulation, which is not conducive to improving the corrosion resistance and adhesion of the substrate.

## 4. Conclusions

Ti/Zr-based synchronous conversion coatings were prepared on 6061 aluminum alloy/7075 aluminum alloy/galvanized steel with different treatment times. We supposed that the formation of synchronous conversion coatings of multi-metal could be roughly divided into three steps by means of OCP measurements.

The corrosion resistance of multi-metal was enhanced markedly after synchronous chemical conversion at 120 s by means of electrochemical PDP and EIS techniques. The surface morphology of the synchronous conversion coatings exhibited uniform and relatively smoothness.

The existence of titanium or zirconium oxides around intermetallic compounds, enrichment and inclusions is also confirmed for providing effective active sites for the nucleation of chemical conversion. Additionally, the Ti/Zr conversion layers on the surface of cathodic $\alpha$-Al (Fe, Mn)Si particles/Cu enrichment/carbon oxides inclusions and the surrounding areas are formed for the first time according to the EPMA measurement. As a result, the coating of 6061 and 7075 aluminum alloy is consisted of metallic oxides ($Al_2O_3$/$TiO_2$/$ZrO_2$) and metal fluorides ($ZrF_4$), while the coating of galvanized steel mainly consists of metallic oxides ($TiO_2$/$Fe_2O_3$/$ZrO_2$). This study provides a significant strategy to improve the corrosion resistance of multi-metal for further application.

**Supplementary Materials:** The following supporting information can be downloaded at: https://www.mdpi.com/article/10.3390/met12122011/s1, Figure S1: Bode and Bode Phase graphs; Table S1: The parameters of $R_s$, $R_f$, $Q_1$, $Q_2$ and $R_{ct}$.

**Author Contributions:** Conceptualization, W.Z.; Data curation, X.L.; Formal analysis, X.L. and Y.L.; Funding acquistion, W.Z.; Investigation, X.L., Y.D. and Y.Z.; Methodology, W.Z, Y.D. and Y.Z.; Project administration, F.X.; Resources, F.T.; Software, X.Q. and Y.L.; Supervision, X.Q., F.X. and F.T.; Visualization, Y.L.; Writing – original draft, W.Z.; Writing—review & editing, W.Z., X.Q. and Y.D. All authors have read and agreed to the published version of the manuscript.

**Funding:** This research was funded by the National Natural Science Foundation of China, Grant Nos. 52075391.

**Conflicts of Interest:** The authors declare no conflict of interest.

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
