# Peer review of "Preparation and Characterization of Synchronous Chemical Conversion Coating on 6061 Aluminum Alloy/7075 Aluminum Alloy/Galvanized Steel Substrates"

_metals, doi:10.3390/met12122011_

Round 1

Reviewer 1 Report

Dear authors,

The main is very interesting. Although a novelty is provided, in my frank opinion, a MAJOR REVISON should be provided before its final publication. The main reasons are described below:

1)    English written (Grammar and Spelling) should be meticulously revised.

2)    The resolutions of the figures are not good. Please change resolutions. Figure 2 is small and does not allow for an adequate analysis; Figure 6 could be with a much better resolution; figure 7 is "compressed" and does not allow visualization of the scales.

3)    Insert sample figures.

4)    Open circuit potential: how many points per decade?

5)    What is the size of the counter electrode?

6)    Into the polarization results, Tafel extrapolation should be clarified.

7)    Show the Bode and Bode Phase graphs. Make a discussion of them. Plot the simulated curve.

8)    Show the number of the time constant by graphs.

The number of time constants can be intimately associated with distinct reactions affecting the electrochemical behavior of the samples examined. An understanding of these numbers of time constants is useful in describing the mechanism of corrosion, intermediate absorbed species and the film of corrosion by-products that constitute a protective barrier, providing transport and diffusion to these species

Take the articles as a reference (Please insert the articles in the bibliographic references of your work):

Distinct heat treatments and powder size ratios affecting mechanical responses of Al/Si/Cu composites (https://journals.sagepub.com/doi/abs/10.1177/00219983211029352), For your Introduction section.

 The Holes of Zn Phosphate and Hot Dip Galvanizing on Electrochemical Behaviors of Multi-Coatings on Steel Substrates (https://www.mdpi.com/2075-4701/12/5/863)

Electrochemical behavior and compressive strength of Al-Cu/xCu composites in NaCl solution (https://link.springer.com/article/10.1007/s10008-020-04890-x)

Osório, W.R.; Peixoto, L.C.; Garcia, A. Electrochemical corrosion behaviour of a Ti-IF steel and a SAE 1020 steel in a 0.5 M NaCl solution. Mater Corros. 2010, (https://onlinelibrary.wiley.com/doi/abs/10.1002/maco.200905420).

 9)    Discuss Nyquist plots.

10) At Experimental section, error ranges for al used dimensions should be indicated.

11) Impedance parameters: when proposing the equivalent circuit, what is the Sum of Sqr. of data between the experimental and simulated curves?

Author Response

Reviewer 1:

The main is very interesting. Although a novelty is provided, in my frank opinion, a MAJOR REVISON should be provided before its final publication. The main reasons are described below:

(1) English written (Grammar and Spelling) should be meticulously revised.

Response: Thank you for your useful suggestion. All writing errors in the new insertions during revision were corrected, including misspellings.

(2) The resolutions of the figures are not good. Please change resolutions. Figure 2 is small and does not allow for an adequate analysis; Figure 6 could be with a much better resolution; figure 7 is "compressed" and does not allow visualization of the scales.

Response: As for resolutions of figures, we accommodated this point in the revised manuscript.

(3) Insert sample figures.

Response: We are very sorry that our samples are all used for dropping experiment and electrochemical experiment, which are destructive experiments

(4) Open circuit potential: how many points per decade?

Response: 15000 points per decade.

(5) What is the size of the counter electrode?

Response:  We use platinum foil electrode, the size is 1 cm x 1 cm.

(6) Into the polarization results, Tafel extrapolation should be clarified.?

Response: The schematic diagram of polarization curve extrapolation method is shown in the figure below

Figure  Schematic Diagram of Polarization Curve Extrapolation Method

(7) Show the Bode and Bode Phase graphs. Make a discussion of them. Plot the simulated curve.

Response: Take 6061 aluminum alloy as an example, the Bode diagram shows that the impedance modulus |Z| of the conversion coating is large. According to the phase diagram, the n of the conversion coating is less than 1, indicating that a defective passive coating is formed on the surface of 6061 aluminum alloy. This paper focuses on the Tafel curves and AC impedance. Bode and Bode phase diagram are not included in the paper, otherwise the structure of the paper will be affected due to too much data.

Bode and Bode Phase graphs of 6061 aluminum alloy

Bode and Bode Phase graphs of conversion film

(8) Show the number of the time constant by graphs.

“The number of time constants can be intimately associated with distinct reactions affecting the electrochemical behavior of the samples examined. An understanding of these numbers of time constants is useful in describing the mechanism of corrosion, intermediate absorbed species and the film of corrosion by-products that constitute a protective barrier, providing transport and diffusion to these species”.

Take the articles as a reference (Please insert the articles in the bibliographic references of your work):

Distinct heat treatments and powder size ratios affecting mechanical responses of Al/Si/Cu composites (https://journals.sagepub.com/doi/abs/10.1177/00219983211029352), For your Introduction section.

The Holes of Zn Phosphate and Hot Dip Galvanizing on Electrochemical Behaviors of Multi-Coatings on Steel Substrates (https://www.mdpi.com/2075-4701/12/5/863).

Electrochemical behavior and compressive strength of Al-Cu/xCu composites in NaCl solution (https://link.springer.com/article/10.1007/s10008-020-04890-x).

Osório, W.R.; Peixoto, L.C.; Garcia, A. Electrochemical corrosion behaviour of a Ti-IF steel and a SAE 1020 steel in a 0.5 M NaCl solution. Mater Corros. 2010, (https://onlinelibrary.wiley.com/doi/abs/10.1002/maco.200905420).

(9) Discuss Nyquist plots.

Response:esistance.

In order to enable an accurate analysis of the impedance diagram, the EIS equivalent circuit reported in Figure 2 is used; Rs is the solution resistance; Rf is the coating resistance; Rct is the charge transfer resistance on the metal surface.   Among them, Rct is one of the most important parameters. Rct of the composite conversion coating is all higher than that of the matrix material, which show that the introduction of conversion coating can significantly reduce the charge transfer on the metal surface.

(10) At Experimental section, error ranges for al used dimensions should be indicated.

Response: The error range of the dropping test is about 0.2-1.0 s, and the error range of electrochemical test is small.

(11) Impedance parameters: when proposing the equivalent circuit, what is the Sum of Sqr. of data between the experimental and simulated curves?

Response: The results of software processing show that the experimental curve is basically consistent with the simulation curve.

Reviewer 2 Report

REPORTS ON: METALS-1988852

The manuscript is very interesting and a great number of experimentations is provided. Although various comments/suggestions are indicated, the manuscript DESERVES its final publication AFTER A MAJOR REVISION, as indicated:

1)    Firstly, the Abstract should be revised and only a simple present tense be used. At least 3 distinctive verbal tenses are used.

2)    In section 2, all values and dimensions should be revised and their corresponding error ranges included.

3)    In line 113, it is mentioned that “...scanning speed was 0.001 V/s.”. Based on this, the follow sentences and references are suggested to be included:

“It is remarked that potential scan rate has an important role in order to minimize the effects of distortion in Tafel slopes and corrosion current density analyses, as previously reported [AA-CC]. However, based on these reports, the adopted 0.001 mV/s has no deleterious effects on those Tafel extrapolations to determine the corrosion current densities of the examined samples.”

[AA] Duarte T, Meyer Y.A. Osório W.R. The Holes of Zn Phosphate and Hot Dip Galvanizing on Electrochemical Behaviors of Multicoatings on Steel Substrates. Metals 2022, 12(5): 863; https://doi.org/10.3390/met12050863

[BB] Zhang X.L., Jiang Zh.H., Yao Zh.P, Song Y., Wu Zh.D. Effects of scan rate on the potentiodynamic polarization curve obtained to determine the Tafel slopes and corrosion current density. Corrosion Science. 2009, 51: 581-587.

[CC] E. McCafferty. Validation of corrosion rates measured by Tafel extrapolation method. Corr. Scie 47 (2005) 3202-3215.

4)    Between lines 117/118, it should be included the number of points per decade was used. Also, if a r.m.s or peak-to-peak was applied.

5)     In line 115, the term “Icorr” should be replaced with “icorr”. This due to the former corresponding with “corrosion rate” and the latter with “corrosion current density”. This modification should be also made in Table 1 and a revision throughout the proposed discussion text concern to matter should also be provided.

6)    In Fig. 1, at least an sample should be selected and, at least, duplicate results be included/depicted.

7)    In Fig. 2(a), (b) and (c), at least Tafel’s extrapolation should be indicated. For this purpose, arrows should be depicted.

8)    Considering Nyquist plots at Figs. 2(d), (e) and (f), the simulated lines should be included. Additionally, the follow sentences and references should included:

“Since an equivalent circuit is used (as shown in Fig. 3), in order to determine the simulated values and compare with experimental data, a CNLS (complex non-linear least squares) simulation is used, as previously reported [AA;DD-GG].”

[DD] Y.A Meyer, RS Bonatti, AD Bortolozo, WR Osório. Electrochemical behavior and compressive strength of Al-Cu/xCu composites in NaCl solution. Journal of Solid State Electrochemistry 25 (2021) 1303-1317.

[EE] YA Meyer, I Menezes, RS Bonatti, AD Bortolozo, WR Osório. EIS investigation of the corrosion behavior of steel bars embedded into modified concretes with eggshell contentes. Metals 202212(3), 417; https://doi.org/10.3390/met12030417

[FF] B. Hirschorn, M. E. Orazem, B. Tribollet, V. Vivier, Isabelle Frateur and M. Musiani. Determination of effective capacitance and film thickness from constant-phase-element parameters. Electrochimica Acta, 55 (2010) 6218-6227.

[GG] B. Hirschorn, M.E. Orazem, B. Tribollet, V. Vivier, I. Frateur and M. Musiani. Constant-Phase-Element Behavior Caused by Resistivity Distributions in Films. J. Electrochem. Soc., 157 (2010) C458-C463.

9)    Table 1 should be revised and error ranges included.

10) Figs. 4(f), 5(f) and 6(f) should be revised due to values are not totalizing 100%.

11)When presenting and discussing Fig. 10, a new sentence should be proposed in order to justify the reason for no selection of a XRD patterns to characterize the resulting phases constituted.

12)It is hardly suggested that the Conclusion section be revised and reworked in order to bullets be included.

______

Author Response

Reviewer 2:

The manuscript is very interesting and a great number of experimentations is provided. Although various comments/suggestions are indicated, the manuscript DESERVES its final publication AFTER A MAJOR REVISION, as indicated:

(1) Firstly, the Abstract should be revised and only a simple present tense be used. At least 3 distinctive verbal tenses are used.

Response: Thank you for your useful suggestion. As Abstract section, we had revised as follows:

“This paper aims to develop synchronous chemical conversion coatings on multi-metal substrates with good corrosion resistance to meet the primer process of new energy light vehicle bodies. Titanium/zirconium-based chemical conversion coatings were prepared on 6061 aluminum alloy/7075 aluminum alloy/galvanized steel substrates. By measuring the open circuit potential (OCP), the formation of a muti-metal synchronous conversion coating can be roughly divided into three steps. Potentiodynamic polarization (PDP) and electrochemical impedance spectroscopy (EIS) techniques showed that the self-corrosion current density of the conversion coating decreased significantly while the resistance increased. The surface morphology and composition of the conversion coatings were observed by scanning electron microscope (SEM) and X-ray photoelectron spectroscopy (XPS). Additionally, the micro-zone characteristics of conversion coatings were analyzed by electron probe microanalyzer (EPMA). The synchronous conversion coatings exhibited uniformity and relative smoothness. Besides, a number of tiny cracks, pores, intermetallic compounds, enrichments and inclusions provided efficient active sites for nucleation of chemical conversion. Consequently, in the synchronous conversion coating, the structure of aluminum alloy mainly consists of Al2O3/TiO2/ZrO2/ZrF4, while the structure of conversion coating of galvanized steel contains TiO2/Fe2O3/ZrO2.”

(2) In section 2, all values and dimensions should be revised and their corresponding error ranges included.

Response: The error range of the dropping test is about 0.2-1.0 s, and the error range of electrochemical test is small. Generally, the data error is very small.

(3) In line 113, it is mentioned that “...scanning speed was 0.001 V/s.”. Based on this, the follow sentences and references are suggested to be included:

“It is remarked that potential scan rate has an important role in order to minimize the effects of distortion in Tafel slopes and corrosion current density analyses, as previously reported [AA-CC]. However, based on these reports, the adopted 0.001 mV/s has no deleterious effects on those Tafel extrapolations to determine the corrosion current densities of the examined samples.”

[AA] Duarte T, Meyer Y.A. Osório W.R. The Holes of Zn Phosphate and Hot Dip Galvanizing on Electrochemical Behaviors of Multicoatings on Steel Substrates. Metals 2022, 12(5): 863; https://doi.org/10.3390/met12050863

[BB] Zhang X.L., Jiang Zh.H., Yao Zh.P, Song Y., Wu Zh.D. Effects of scan rate on the potentiodynamic polarization curve obtained to determine the Tafel slopes and corrosion current density. Corrosion Science. 2009, 51: 581-587.

[CC] E. McCafferty. Validation of corrosion rates measured by Tafel extrapolation method. Corr. Scie 47 (2005) 3202-3215.

Response: Thank you for your suggestion. For the contents of line 113, we had revised as follows:

“It is remarked that potential scan rate has an important role in order to minimize the effects of distortion in Tafel slopes and corrosion current density analyses, as previously reported [27-28]. However, based on these reports, the adopted 0.001 mV/s has no deleterious effects on those Tafel extrapolations to determine the corrosion current densi-ties of the examined samples.” These changes are presented in red font from (Page 3, line 113-117) of manuscript.

(4) Between lines 117/118, it should be included the number of points per decade was used. Also, if a r.m.s or peak-to-peak was applied.

Response: Almost 1000 points per decade was used. Generally speaking, Tafel extrapolation should be adopted.

(5) In line 115, the term “Icorr” should be replaced with “icorr”. This due to the former corresponding with “corrosion rate” and the latter with “corrosion current density”. This modification should be also made in Table 1 and a revision throughout the proposed discussion text concern to matter should also be provided.

Response: We changed “Icorr” to “icorr” in the manuscript.

(6) In Fig. 1, at least an sample should be selected and, at least, duplicate results be included/depicted.

Response: It can be seen from the Fig. 1, there are three distinct phases of all coating formation. Take 7075 aluminum alloy as an example, the OCP of 7075 aluminum alloy dropped rapidly to more negative values within the first 93.4 s. At this stage, the surface of the sample begins to nucleate, but the film formation rate is far less than the dissolution rate of natural oxide film, so the potential drops rapidly. After the removal of natural multi-metal oxide, hydrogen evolution and oxygen reduction reactions, due to the increase of local alkalinity, synchronous chemical conversion coatings starts to form on cathodic areas of the substrate, afterwards growing in a lateral direction until completely covering the substrate. The maximum in the OCP curves of 7075 aluminum alloy is 119.9 s, which was taken as the optimal conversion time. The complete formation of  chemical conversion coatings is after 200 s, and the coatings tend to grow further.

(7) In Fig. 2(a), (b) and (c), at least Tafel’s extrapolation should be indicated. For this purpose, arrows should be depicted.

Response: Response: The schematic diagram of polarization curve extrapolation method is shown in the figure below, all polarization curves are fitted using the above method.

Figure  Schematic Diagram of Polarization Curve Extrapolation Method

(8) Considering Nyquist plots at Figs. 2(d), (e) and (f), the simulated lines should be included. Additionally, the follow sentences and references should included:

“Since an equivalent circuit is used (as shown in Fig. 3), in order to determine the simulated values and compare with experimental data, a CNLS (complex non-linear least squares) simulation is used, as previously reported [AA;DD-GG].”

[DD] Y.A Meyer, RS Bonatti, AD Bortolozo, WR Osório. Electrochemical behavior and compressive strength of Al-Cu/xCu composites in NaCl solution. Journal of Solid State Electrochemistry 25 (2021) 1303-1317.

[EE] YA Meyer, I Menezes, RS Bonatti, AD Bortolozo, WR Osório. EIS investigation of the corrosion behavior of steel bars embedded into modified concretes with eggshell contentes. Metals 2022, 12(3), 417; https://doi.org/10.3390/met12030417

[FF] B. Hirschorn, M. E. Orazem, B. Tribollet, V. Vivier, Isabelle Frateur and M. Musiani. Determination of effective capacitance and film thickness from constant-phase-element parameters. Electrochimica Acta, 55 (2010) 6218-6227.

[GG] B. Hirschorn, M.E. Orazem, B. Tribollet, V. Vivier, I. Frateur and M. Musiani. Constant-Phase-Element Behavior Caused by Resistivity Distributions in Films. J. Electrochem. Soc., 157 (2010) C458-C463.

Response: We redraw the Nyquist plots of Fig. 2(d), (e) and (f), including the simulation lines. Meanwhile, the manuscript adds the relevant sentence content as follows:

“Since an equivalent circuit is used (as shown in Figure 3), in order to determine the simulated values and compare with experimental data, a CNLS (complex non-linear least squares) simulation is used, as previously reported [29-31]. These changes are presented in red font from (Page 4, line 165-167) of manuscript.

(9) Table 1 should be revised and error ranges included.

Response: The error range of the dropping test is about 0.2-1.0 s, and the error range of electrochemical test is small. Generally, the data error is very small. Therefore, the error is not indicated in Table 1.

(10) Figs. 4(f), 5(f) and 6(f) should be revised due to values are not totalizing 100%.

Response: Here, it is mainly to explain the morphology, not the element content. Therefore, we delete the element topography

(11) When presenting and discussing Fig. 10, a new sentence should be proposed in order to justify the reason for no selection of a XRD patterns to characterize the resulting phases constituted.

Response:The thickness of the coating is too thin, so the XRD results can not well characterize the phase of the coating on the surface.

(12) It is hardly suggested that the Conclusion section be revised and reworked in order to bullets be included.

Response: As suggested, I have revised it in the conclusion

Reviewer 3 Report

Abstract

Why is the word "self" in self-corrosion current density? the corrosion current density is obtained after polarization, unlike the OCP which is not polarized. Please keep it as corrosion current density.

Introduction
1. Include the reference(s) for the discussion in lines 32 to 37

Methods and Materials
1. it is not clear how the conversion coating was produced. Is it during the OCP in 3.5 %NaCl solution (for 30min) or the immersion in H2TiF6, H2ZrF6  solutions for 30 min.

Results

1. Figure one is not up 30 min (1800s)?

2. How can the authors substantiate the assumptions made that "there are three distinct phases of all coating formation"? 

3. Figure 2 is not that legible. 

4. Please provide the Bode plot as well (Modulus and Phase angle plots)

5. Provide the morphology of the coatings before corrosion

6. Table 1 does not contain all elements (Rs, Rf, Q1, Q2 ????) shown in the circuit in Figure 3.

7. The SEM images in Figures 7-9 are distorted and not clear 

Author Response

Reviewer 3:

Abstract:

(1) Why is the word "self" in self-corrosion current density? the corrosion current density is obtained after polarization, unlike the OCP which is not polarized. Please keep it as corrosion current density.

Response: Polarization curve extrapolation method is used to to calculate the icorr and Ecorr. icorr represents the corrosion rate in natural state, which has been agreed to become self corrosion current. OCP mainly represents the relationship between potential and time. 

Introduction:

(1) Include the reference(s) for the discussion in lines 32 to 37.

Response: This part mainly comes from news reports and public materials of related companies.

Methods and Materials:

(1) it is not clear how the conversion coating was produced. Is it during the OCP in 3.5 %NaCl solution (for 30min) or the immersion in H2TiF6, H2ZrF6 solutions for 30 min.

Response: The conversion coating is prepared in H2TiF6, H2ZrF6 solutions for immersion of 30 min.

Results:

(1) Figure one is not up 30 min (1800s)?

Response: I'm very sorry. It's wrong. It's 25min. It has been modified in the paper. 

(2) How can the authors substantiate the assumptions made that "there are three distinct phases of all coating formation"?

Response:According to the shape of OCP curve and the growth characteristics of the coating, the corresponding inference is made.

(3) Figure 2 is not that legible.

Response: We have corrected Figure 2 with a much better resolution in the revised manuscript.

(4) Please provide the Bode plot as well (Modulus and Phase angle plots).

Response: Take 6061 aluminum alloy as an example, the Bode diagram shows that the impedance modulus |Z| of the conversion coating is large. According to the phase diagram, the n of the conversion coating is less than 1, indicating that a defective passive coating is formed on the surface of 6061 aluminum alloy. This paper focuses on the Tafel curves and AC impedance. Bode and Bode phase diagram are not included in the paper, otherwise the structure of the paper will be affected due to too much data.

Bode and Bode Phase graphs of 6061 aluminum alloy

Bode and Bode Phase graphs of conversion film

(5) Provide the morphology of the coatings before corrosion.

Response: I'm sorry, as a comparison, the original sample was only subjected to relevant performance tests and its surface morphology was not observed.

(6) Table 1 does not contain all elements (Rs, Rf, Q1, Q2 ????) shown in the circuit in Figure 3.

Response: In order to enable an accurate analysis of the impedance diagram, the EIS equivalent circuit reported in Figure 2 is used, and the EIS equivalent circuit contain the parameters of Rs, Rf, Q1, Q2 and Rct.   Among them, Rct is one of the most important parameters, which can characterize the charge transfer on the metal surface. Therefore, this paper only discuss the parameter of Rct.

(7) The SEM images in Figures 7-9 are distorted and not clear.

Response: We accommodated this point in the revised manuscript (Revised Figures 7-9).

Reviewer 4 Report

Paper Preparation and characterization of synchronous chemical conversion coating on 6061 aluminum alloy/7075 aluminum alloy/galvanized steel substrates present some interesting experimental results in a requested field with a possible new technology for coatings. The subject proposed is interesting and proper presented using new techniques (XPS and EMPA) and consecrated techniques like SEM, EDS or EIS. The results are representative and important for the field. 

line 56: Alodine2840 - Alodine 2840 

re-structure this phrase: Electrical equivalent circuits of without and with synchronous chemical conversion of multi-metal samples

in figure 4  b) is an intermetallic compound or an oxide (ceramic nature) ? 

how many : EDS result of the conversion coating  were made , provide an average value and a standard deviation (StDev) - the same for f) result from figures 5 and 6 

better quality of figure 7 is required 

line 274: delete a dot : ..

line 344: use upper script for F−

re-structure the conclusions 

 the references Section is well prepared and contain many important titales for this field 

Author Response

Reviewer 4:

Paper Preparation and characterization of synchronous chemical conversion coating on 6061 aluminum alloy/7075 aluminum alloy/galvanized steel substrates present some interesting experimental results in a requested field with a possible new technology for coatings. The subject proposed is interesting and proper presented using new techniques (XPS and EMPA) and consecrated techniques like SEM, EDS or EIS. The results are representative and important for the field.

(1) line 56: Alodine2840 - Alodine 2840

re-structure this phrase: Electrical equivalent circuits of without and with synchronous chemical conversion of multi-metal samples.

Response: We changed “Electrical equivalent circuits of without and with synchronous chemical conversion of multi-metal samples.” to “Equivalent circuits for untreated and simultaneously chemical converted treated multi-metal samples.” in the manuscript. These changes are presented in red font from (Page 6, line 201) of manuscript.

(2) in figure 4 (b) is an intermetallic compound or an oxide (ceramic nature) ?

Response: Chemical compound in figure 4 (b) is an intermetallic compound.

(3) how many: EDS result of the conversion coating were made, provide an average value and a standard deviation (StDev) - the same for f) result from figures 5 and 6.

Response: Here, it is mainly to explain the morphology, not the element content. Therefore, we delete the element topography

(4) better quality of figure 7 is required

Response: We accommodated this point in the revised manuscript (Revised Figure. 7).

(5) line 274: delete a dot : ..

Response: We corrected this error in the revised manuscript.

(6) line 344: use upper script for F

Response: We changed “F-” to “Fluoride Ion” in the manuscript (Page 13, line 351)

(7) re-structure the conclusions, the references Section is well prepared and contain many important titales for this field.

Response: As Conclusions section, we had revised as follows:

“Ti/Zr based synchronous conversion coatings were prepared on 6061 aluminum alloy/7075 aluminum alloy/galvanized steel with different treatment times. We sup-posed that the formation of synchronous conversion coatings of multi-metal could be roughly divided into three steps by means of OCP measurements. The corrosion re-sistance of multi-metal was enhanced markedly after synchronous chemical conversion at 120 s by means of electrochemical PDP and EIS techniques. The surface morphology of the synchronous conversion coatings exhibited uniform and relatively smoothness. The existence of titanium or zirconium oxides around intermetallic compounds, en-richment and inclusions, is also confirmed for providing effective active sites for the nucleation of chemical conversion. Additionally, the Ti/Zr conversion layers on the surface of cathodic α-Al (Fe, Mn)Si particles/Cu enrichment/carbon oxides inclusions and the surrounding areas are formed for the first time according to the EPMA meas-urement. As a result, the coating of 6061 and 7075 aluminum alloy is consisted of me-tallic oxides(Al2O3/TiO2/ZrO2) and metal fluorides(ZrF4), while the coating of galvanized steel mainly consists of metallic oxides(TiO2/Fe2O3/ZrO2). This study provides a signif-icant strategy to improve the corrosion resistance of multi-metal for further application.”

References

27  Duarte T, Meyer Y.A. Osório W.R. The Holes of Zn Phosphate and Hot Dip Galvanizing on Electrochemical Behaviors of Multicoatings on Steel Substrates. Metals 2022, 12(5): 863; https://doi.org/10.3390/met12050863

28  Zhang X.L.; Jiang Zh.H.; Yao Zh.P.; Song Y.; Wu Zh.D.; Effects of scan rate on the potentiodynamic polarization curve obtained to determine the Tafel slopes and corrosion current density. Corrosion Science. 2009, 51: 581-587.

29  E McCafferty.; Validation of corrosion rates measured by the Tafel extrapolation method. Corrosion science, 2005, 47(12): 3202-3215.

30  Y.A Meyer.; RS Bonatti.; AD Bortolozo.; WR Osório.; Electrochemical behavior and compressive strength of Al-Cu/xCu composites in NaCl solution. Journal of Solid State Electrochemistry. 2021 25: 1303-1317.

31  YA Meyer.; I Menezes.; RS Bonatti.; AD Bortolozo.; WR Osório.; EIS investigation of the corrosion behavior of steel bars embedded into modified concretes with eggshell contentes. Metals. 2022, 12(3), 417.

32  B. Hirschorn.; M. E. Orazem.; B. Tribollet.; V. Vivier.; Isabelle Frateur.; M. Musiani. Determination of effective capacitance and film thickness from constant-phase-element parameters. Electrochimica Acta. 2010 55, 6218-6227.

33 B. Hirschorn, M.E. Orazem, B. Tribollet, V. Vivier, I. Frateur and M. Musiani. Constant-Phase-Element Behavior Caused by Resistivity Distributions in Films. J. Electrochem. Soc.. 2010,157, C458-C463.

Round 2

Reviewer 1 Report

The authors submitted an acceptable version for publication after the reviewer's comments.

Author Response

We thank the reviewers for their valuable and insightful comments.

Reviewer 2 Report

After corrections/ modifications, the manuscript deserves its publication.

Author Response

(The authors gave the same response as above.)

Reviewer 3 Report

(1) Include the reference(s) for the discussion in lines 32 to 37 even if it comes from the news or public sources, website, company white paper, etc are references. it is not sufficient to make a claim which is not traceable.

(2) How can the authors substantiate the assumptions made that "there are three distinct phases of all coating formation"? Does the OCP has any relationship with the growth of the coating?

(3) the authors stated in the materials and methods: "The tests were 108 carried out in 3.5 mass% NaCl solution for 25 min to get a steady open circuit potential 109 (OCP), using the standard three-electrode cell equipped with the coated sample (1 cm x 1 110 cm) as a working electrode," while in subsection 3.1 they stated "The formation of synchronous chemical conversion were followed by measuring the 136 OCP of 6061 aluminum alloy/7075 aluminum alloy/galvanized steel samples immersed in 137 a conversion bath of H2TiF6 2.2 ml·L-1 , H2ZrF6 1 ml·L-1 (Figure 1)." In what media was the OCP measured?

(4) The authors stated that "This paper focuses on the Tafel curves and AC impedance. Bode and Bode phase diagram are not included in the paper, otherwise the structure of the paper will be affected due to too much data."

The Bode plots is just an integral part of the AC Impedance. If the manuscript will be affected by to many data, please include them as supplementary materials.

(5) Since the EIS equivalent circuit is shown with the parameters in the manuscript, it is imperative that the parameters of Rs, Rf, Q1, Q2 and Rct are presented as well. At least include them as a supplementary material

(6) the conclusion: "We supposed 357 that the formation of synchronous conversion coatings of multi-metal could be roughly 358 divided into three steps by means of OCP measurements." is not based on substantial outcome of this study. Note that during the OCP the system has not attain equilibrium especially when it is just for 25 min. How can the authors justify an intrinsic property by a quasi-stable or unstable OCP?

Author Response

Reviewer 3:

  1. Include the reference(s) for the discussion in lines 32 to 37 even if it comes from the news or public sources, website, company white paper, etc are references. it is not sufficient to make a claim which is not traceable.

Response:Thank you for the tip. This paper mainly analyzes and discusses the current situation of new energy vehicle surface treatment technology. As for the expression in No. 32-37 mentioned by you, there is basically a consensus on the use of body materials in the automobile industry. Please kindly understand.

  1. 2. How can the authors substantiate the assumptions made that "there are three distinct phases of all coating formation"? Does the OCP has any relationship with the growth of the coating?

Response:Determine the stage of coating growth according to the shape of OCP curve (change of potential ). OCP can reflect the change of potential during coating growth, thus further affecting the coating growth state.

  1. 3. the authors stated in the materials and methods: "The tests were carried out in 3.5 mass% NaCl solution for 25 min to get a steady open circuit potential (OCP), using the standard three-electrode cell equipped with the coated sample (1 cm x 1 110 cm) as a working electrode," while in subsection 3.1 they stated "The formation of synchronous chemical conversion were followed by measuring the OCP of 6061 aluminum alloy/7075 aluminum alloy/galvanized steel samples immersed in a conversion bath of H2TiF62 ml·L-1 , H2ZrF61 ml·L-1 (Figure 1)." In what media was the OCP measured?

Response:OCP is a test method in electrochemistry. When immersing the working electrode in a concentration of 3.5% NaCl for AC impedance experiment, its open circuit potential should be measured first. When the open circuit potential is measured in the conversion solution or electrolyte of the working electrode coating, it is mainly used to judge the potential step process, so as to reverse the coating formation process. Therefore, the OCP mentioned in the experimental part of this article is the condition used to measure the AC impedance first. It is mentioned later that the measurement of OCP in the conversion solution measured in a conversion bath of H2TiF6 2.2 ml·L-1 , H2ZrF6 1 ml·L-1 is a judgment for the formation of the coating.

  1. 4. The authors stated that "This paper focuses on the Tafel curves and AC impedance. Bode and Bode phase diagram are not included in the paper, otherwise the structure of the paper will be affected due to too much data."

The Bode plots is just an integral part of the AC Impedance. If the manuscript will be affected by to many data, please include them as supplementary materials.

Response:Take 6061 aluminum alloy as an example, the Bode diagram shows that the impedance modulus |Z| of the conversion coating is large. According to the phase diagram, the n of the conversion coating is less than 1, indicating that a defective passive coating is formed on the surface of 6061 aluminum alloy. The Bode and Bode Phase graphs is shown in Figure S1 of Supplementary Material.

Figure S1. Bode and Bode Phase graphs

  1. Since the EIS equivalent circuit is shown with the parameters in the manuscript, it is imperative that the parameters of Rs, Rf, Q1, Q2 and Rct are presented as well. At least include them as a supplementary material

Response:The parameters of Rs, Rf, Q1, Q2 and Rct is shown in Table S1 of the Supplementary Material. 

Table S1. The parameters of Rs, Rf, Q1, Q2 and Rct

Samples

icorr/μA·cm-2

Ecorr/V

Rct/Ω

 Rs/Ω

 Rf/Ω

 Q1  

Q2

with conversion AA6061

0.174

-1.084

84610

8.66

3675

6.317×10-5

9.334×10-5

without conversion AA6061

1.096

-1.151

0.268

12.97

919

8.661×10-4

6.221×10-4

with conversion AA7075

0.018

-0.603

23250

7.03

2876

9.112×10-5

3.666×10-5

without conversion AA7075

1.470

-0.653

19330

11.64

799

7.663×10-4

2.729×10-4

with conversion galvanized steel

1.012

-0.838

13010

6.07

2017

1.117×10-4

3.915×10-4

without conversion galvanized steel

6.312

-0.938

5441

9.88

366

7.664×10-4

9.226×10-3

  1. 6. the conclusion: "We supposed that the formation of synchronous conversion coatings of multi-metal could be roughly divided into three steps by means of OCP measurements." is not based on substantial outcome of this study. Note that during the OCP the system has not attain equilibrium especially when it is just for 25 min. How can the authors justify an intrinsic property by a quasi-stable or unstable OCP?

Response:25 min is enough for the deposition of conversion film. At present, the formation time of most conversion films is between 1 to 20min. Therefore, OCP of 25 min can describe the formation of synchronous conversion coatings of multi-metal.

Round 3

Reviewer 3 Report

1. Can the authors unify the frequency range in the Bode plots provided in the supplementary material?

2. Also, they should make a reference to the supplementary material in the manuscript.

3. The clarification regarding the OCP is still not sufficient, it is still ambiguous what the authors intended. 

Highlight all changes in the manuscript.

Author Response

Dear editor and reviewers

    Thank you for your comments on this article. Based on the opinions of all experts, my point-to-point summary and reply are as follows:

  1. Can the authors unify the frequency range in the Bode plots provided in the supplementary material?

Response:We have unified the frequency range in the Bode plots in the supplementary material.

  1. Also, they should make a reference to the supplementary material in the manuscript.

Response:We add a sentence“The relevant analysis on Bode and Bode Phase were seen in the supplementary material ”in the manuscript.

  1. The clarification regarding the OCP is still not sufficient, it is still ambiguous what the authors intended. Highlight all changes in the manuscript.

Response:We added the purpose of OCP experiment in the manuscript. The OCP test is carried out in the conversion solution, and the formation process of the conversion film is judged by its potential change. 
